The application of deep learning in early enamel demineralization detection

He Ketai 1
Zhang Rongxiu 2
Liang Muchun 1
Tian Keyue 1
Luo Kaihui 1
Chen Ruoshi 3
Ren Jianpeng 3
Wang Jiajun 4
Li Juan 1 lijuan@scu.edu.cn
1 State Key Laboratory of Oral Diseases & National Center for Stomatology & National Clinical Research Center for Oral Diseases, West China Hospital of Stomatology, Sichuan University , Chengdu , China
2 Department of Stomatology, The First Affiliated Hospital of Bengbu Medical University , Chengdu , China
3 Chengdu Boltzmann Intelligence Technology Co., Ltd , Chengdu , China
4 College of Software, Sichuan University , Chengdu , China
Kuijpers-Jagtman Anne Marie
Electronic publication date: 2025 Jan 2
Publication date: 2025
Volume: 13
Electronic Location ID: e18593
Received 2024 May 10; Accepted 2024 Nov 5
Copyright: © 2025 He et al.
Copyright year: 2025
Copyright holder: He et al.
License: This is an open access article distributed under the terms of the Creative Commons Attribution License, which permits unrestricted use, distribution, reproduction and adaptation in any medium and for any purpose provided that it is properly attributed. For attribution, the original author(s), title, publication source (PeerJ) and either DOI or URL of the article must be cited.
License URL: https://creativecommons.org/licenses/by/4.0/

Keywords: Artificial intelligence, Diagnostic imaging, Deep learning, Tooth demineralization

Funding: National Natural Science Foundation 32271364 and 31971240 Major Special Science and Technology Project of Sichuan Province 2022ZDZX0031 Angelalign Scientific Research Fund SDTS21–4–01 Clinical Research Project of West China Hospital of Stomatology, Sichuan University LCYJ2019-22 This study was funded by the National Natural Science Foundation (32271364) and (31971240), the Major Special Science and Technology Project of Sichuan Province (grant no. 2022ZDZX0031), the Angelalign Scientific Research Fund (grant number SDTS21–4–01), and the Clinical Research Project of West China Hospital of Stomatology, Sichuan University (LCYJ2019-22). The funders had no role in study design, data collection and analysis, decision to publish, or preparation of the manuscript.

==============================
Objective

The study aims to develop a diagnostic model using intraoral photographs to accurately detect and classify early detection of enamel demineralization on tooth surfaces.

Methods

A retrospective analysis was conducted with 208 patients aged 14 to 44. A total of 624 high-quality digital images captured under standardized conditions were used to construct a deep learning model based on the Mask region-based convolutional neural network (Mask R-CNN). The model was trained to automate the detection of enamel demineralization. Its performance was compared to two junior dentists’ diagnostic abilities.

Results

The model achieved an F1-score of 0.856 for detecting demineralized teeth on the validation set, a metric that reflects comprehensive diagnostic performance, demonstrating performance close to that of senior dentists. With the the model’s assistance, the junior dentists’ average F1-scores improved significantly—from 0.713 and 0.689 to 0.897 and 0.949, respectively (p < 0.05). The model accurately segmented tooth surfaces and detected demineralized areas, allowing for precise detection of demineralized areas and monitoring of lesion progression.

Conclusion

Deep learning can accurately segment tooth surfaces and lesion contours, enhancing the precision, accuracy, and efficiency of enamel demineralization diagnosis and area delineation.

Introduction

Enamel demineralization occurs when acidic conditions in the mouth such as bacterial acids or acidic diets dissolve enamel minerals like calcium and phosphate (Besnard et al., 2024; Cazzolla et al., 2018). The refractive index of affected enamel changes due to this process, resulting in the formation of white spot lesions (WSL) (Lazar et al., 2023). Classification based on area for demineralization is as follows (Benkaddour et al., 2014): Score 1—no WSL; Score 2—slight WSL; Score 3—severe WSL; Score 4—WSL with caries. However, particularly during orthodontic treatment, the progression of demineralization in some patients may be very subtle and slow, even though it poses a threat to dental health. Various degrees of demineralization not only impact the aesthetic orthodontic results (Tasios et al., 2019) and the tooth structure (Mummolo et al., 2020) but may also lead to caries (Jasso-Ruiz et al., 2020).

Beyond grading challenges and subtle progression, its clinical detection is also cumbersome and inefficient, which further affects the prognosis of the disease. The gold standard is transverse microradiography (TMR), but it is an invasive examination and not commonly used in clinical practice (Fernando et al., 2022). Clinical methods mainly rely on evidence-based approaches, such as visual examination combined with probing, occasionally employing staining agents and imaging examinations (Benkaddour et al., 2014). Senior dentists conducting clinical examinations and combining them with patient histories can achieve high recognition rates and serve as a primary reference standard in specific situations (Cömlekoğlu et al., 2013). However, this approach demands a high level of clinical experience from the examiner. Monitoring demineralization progression after orthodontic treatment is time-consuming and difficult to detect visually unless significant advancement occurs.

Therefore, utilizing deep learning (DL) to develop a precise and automated lesion identification and grading system is beneficial for continuous monitoring of dental conditions, preventing the appearance of demineralization. In recent years, DL has been used in many areas of dentistry and has achieved excellent results, including caries detection, orthodontic markers, implant system, plaque division, periodontal status recognition, and dental position classification (Fatima et al., 2022). Notably, it has been used effectively in caries diagnosis through X-rays (Li et al., 2022) and digital photographs taken with cameras and smartphones (Estai et al., 2016; Zhang et al., 2020). Recent studies have also attempted various imaging methods to detect or diagnose early caries (Walsh et al., 2021).

To quantify the extent and monitor the real-time progress of demineralization, we developed an accurate and efficient lesion identification model based on intraoral photographs. The model quantifies the area of demineralization and assists in the interception and treatment. We utilized digital photographs as the training dataset and employed the improved Mask region-based convolutional neural network (Mask R-CNN) network for demineralized area segmentation. Our assessment shows that the model can accurately and efficiently detect demineralized areas, enhancing the diagnostic accuracy of junior dentists and bringing them closer to the level of senior dentists.

Materials and Methods

Trial design

All procedures involving digital photographs of human subjects received approval from the West China Hospital of Stomatology Ethics Committee (WCHSIRB-CT-2022-044). Only patients who consented to the use of their intraoral images for research were included in this study. The photographs were taken by a nurse specializing in oral photography. A total of 237 adult patients who underwent oral examinations at a single institution, from January to December 2022, were selected. Each patient had one digital photograph taken in the frontal occlusal position, one in the left occlusal position, and one in the right occlusal position. The photographs were divided into training, validation, and test sets in a ratio of 7:2:1. Two senior dentists reached a consensus through clinical examinations and patient history discussions, marking tooth contours and demineralized lesions on the photos, which served as the required samples for the study. Dental photos were taken using a professional DSLR camera, using macro flash after tooth cleaning and drying (Kühnisch et al., 2022). All images were stored in jpeg format with RGB colour for use in this research project.

Inclusion criteria for photos: (1) permanent dentition; (2) presence of only early enamel demineralization issues; (3) all three images clear and unblocked. Exclusion criteria: (1) obscured dentition; (2) extensive tooth loss or previous treatments like dental restorations or implants.

Table 1 displays the demographic and clinical characteristics of the patients.

Table 1 Manual discrimination of demineralization presence and image quality.

Characteristic	Training set	Validation set	Testing set	
Age, range	14–47	14–41	14–34	
Sex, n (%)				
Male	29.2	32.5	25.0	
Female	70.7	67.5	75.0	
Whether there is enamel demineralization	
Yes	96	29	12	
No	52	11	8	

The dental photos were retrospectively and randomly used for enamel demineralization detection by dentists with or without the DL assistance. Two senior dentists (Juan Li, Rongxiu Zhang), two junior dentists (Ketai He, Keyue Tian), and one computer scientist(Jiajun Wang) participated in the experiment. The computer scientist created a DL model based on the annotated dataset, the two senior dentists supplied example annotations. The two junior dentists subjectively assessed the photos with and without the assistance of DL. Finally, the recognition of two junior dentists without DL assistance was compared with that with DL assistance.

An overview of the study is presented in Fig. 1.

Figure 1 Study flowchart.

The demineralization index, representing the percentage of demineralized area relative to the total tooth area, was calculated using DL on the training set. The maxillary and mandibular anterior teeth were calculated in the frontal occlusal photos and the respective posterior teeth (excluding the second and third molars) were calculated in the lateral photos on each side.

Sample annotation

Labelme is a Python-based image annotation tool with a graphical interface built using Qt application framework. We utilized version 4.5.13 to annotate tooth contours, marking early enamel lesions when present, and outputting annotations in JavaScript Object Notation (JSON), a lightweight data-interchange format.

In our study, two senior dentists with over 10 years of experience examined patients and conducted visual inspections based on their medical histories. Subsequently, they marked tooth contours and corresponding demineralized areas on digital photos. The annotation rules are outlined in Table 2, and an illustration is provided in Fig. 2. The dentists were blinded to each other’s responses. To minimize potential subjective impacts on the results, the two dentists underwent testing before the experiment, resulting in a Cohen’s κ > 0.8(0.858), indicating consistency. Only images with consistent annotations from both senior dentists were included in the database, which is critical for the study’s validity.

Table 2 Annotation rules.

Character	Definition	
Group ID*		
P	Frontal	
R	Right	
L	Left	
Added orientation		
T	Tooth surface	
W	WSL	
Note:

*Annotation rules: Indicate the direction represented by the marked surface: P, Frontal; R, Right; L, Left; Annotation points are labelled as “Direction represented (P\R\L) + Tooth position (International Tooth Numbering System) + Type of annotation (T\W) [+ serial number]”–marking premolars and molars in lateral occlusal views and incisors, canines, and premolars in frontal views.

Figure 2 Annotation example.

Red represents P 11 T, green represents P 21 T, and brown represents P 11 W 1.

Preprocessing

Out of the initial 711 digital photos, 87 were excluded. Ultimately, a total of 624 digital photos were included in the study. The digital photographs were randomly divided into three groups: training, testing, and validation. The data was divided in a ratio of 7:2:1, ensuring that each patient’s photos belonged to only one subset. For the purposes of training, validating, and testing the model, the input picture resolution was standardized. The training set comprised 444 images from 148 patients. The validation set included 120 images of 40 patients, and the test set had a total of 60 photos of 20 patients for testing.

DL model for enamel demineralization detection and area calculation

Data augmentation is a data-space solution to address the limited data problem. Current deep convolutional neural networks rely on large datasets to avoid overfitting. In the medical image analysis domain, obtaining a substantial number of similar images is challenging due to ethical constraints. Therefore, data augmentation techniques that enhance dataset size and quality become a viable solution. When processing the secondary database, the team employed three data augmentation methods: cutout, cutmix, and mixup. Additionally, the project utilized an improved Mask R-CNN network for demineralized area segmentation, with the ConvNext network serving as the backbone for the enhanced Mask R-CNN network. Subsequently, the segmented demineralized area was used as input for the Segment Anything network to segment teeth, ensuring precise tooth segmentation and area calculation. This method significantly enhanced recognition efficiency. Pixel area calculations were then used to determine the percentage of the demineralized area relative to the total tooth area, providing input for subsequent scoring.

The team conducted area calculations based on the segmented mask information, and the results were automatically recorded in the experimental records (e.g., Excel spreadsheet).

We call the result of this calculation the demineralized index. In other words, the index is the demineralization area divided by the total tooth surface area. It can reflect the proportion of tooth demineralization, and thus more accurately reflect the severity of tooth enamel demineralization. This index was calculated for specific tooth positions in the training set.

Statistical analysis and evaluation criteria

Excel and SPSS 26.0 were used for all statistical analyses. We assessed the strategy’s effectiveness using metrics like sensitivity (SEN), specificity (SPEC), F1-score, positive predictive value (PPV), negative predictive value (NPV), receiver operating characteristic (ROC) curves, and area under the ROC curve (AUC). These measures were computed for each individual, each picture, and each tooth (Li et al., 2022). The following formulas was used to calculate the metrics:

NPV=TNTN+FN

PPV=TPTP+FP

SEN=TPTP+FN

SPEC=TNTN+FP

F1-score=2∗PPV∗SENPPV+SEN

where TP, FP, TN, and FN represent true positive, false positive, true negative, and false negative, respectively. Positive/negative indicates the model predicting the presence/absence of demineralization, and true/false indicates correct/incorrect predictions. The F1-score, representing the harmonic mean of PPV and SEN, is a commonly used evaluation metric for classification models in DL. It balances these two metrics and is a comprehensive measure of model performance, especially when there is an imbalance between positive and negative samples. In this study, F1-score was used to evaluate the accuracy and reliability of the DL model and junior dentists in detecting demineralization. Perfect PPV and SEN are indicated by the maximum F1-score of 1, which is 1, and zero for either PPV or SEN. The range of all metric values is 0 to 1 (Li et al., 2022).

Subjective assessment and DL-assisted evaluation

The DL model underwent evaluation on a validation set comprising 120 images from 40 patients. To compare the performance under the guidance of DL, two junior dentists with 3 years of experience independently assessed the same set of 60 images from 20 patients, and the DL model also assessed this test set. These junior dentists relied solely on their clinical experience without additional training. Both junior dentists underwent a consistency assessment before the experiment (Cohen’s κ > 0.8). For these photos, they assessed the concordance between the DL screening outcomes and their subjective visual evaluations. If the results were inconsistent, the dentists made final decisions based on their judgments, referring to the DL’s results (Li et al., 2022).

Experimental setup

Experiments were conducted using Python 3.8 and PyTorch 1.10 on a single NVIDIA V100 GPU to ensure sufficient computing power to handle DL tasks. The ConvNext-T network was used as the backbone for the Mask R-CNN network for demineralized area segmentation. The ViT-H SAM network served as the tooth area segmentation network. Before training, all images were standardized to ensure consistent data distribution. During training, we employed five-fold cross-validation to obtain better model parameters. Regarding hyperparameter configuration, we used the SGD optimizer with an initial momentum parameter of 0.9, weight decay parameter of 0.0001, learning rate of 0.004, batch size of 8, and a total of 30 training epochs. The model achieved optimal performance at the end of the 26th epoch. This model is then used on the test set.

Results

Model performance

On the validation set, the model demonstrated a SEN at 0.935, SPEC at 0.789, PPV at 0.968, NPV at 0.991, and an F1-score at 0.856 at the tooth level (refer to Table 3). The model’s ROC curves, depicting AUC values at the individual, image, and tooth levels, are presented in Fig. 3. On the test set, the model also showed a strong capacity to learn features from digitally annotated photos. Compared to the junior dentists, it performed well in detecting demineralization (see Table 4). Specifically, for different tooth positions, the model performed best in recognizing upper premolars, with F1-scores of 0.963 on the test sets. The lowest performance was observed in recognizing lower molars, with F1-scores of 0.727 (see Table S3 for detailed dentition test results). Figure 4 illustrates the DL model’s performance in detecting enamel demineralization.

Table 3 DL’s performance compared to the gold standard in the validation set (categorized by Individual, Image, and Tooth).

	TP	FP	TN	FN	NPV	PPV	SEN	SPEC	F1-score	
Individual level1	29	2	9	0	1.000	0.935	1.000	0.818	0.967	
Image level2	63	7	46	4	0.920	0.900	0.940	0.868	0.920	
Tooth level3	101	27	810	7	0.991	0.789	0.935	0.968	0.856	
Notes:

1 Individual level: Assessment based on each patient.

2 Image level: Evaluation considering three images to determine the presence of demineralization in a patient.

3 Tooth level: Assessment focusing on whether a specific tooth exhibits demineralization in a single image.

Figure 3 ROC curves associated with the DL model.

AUCs are provided in parentheses.

Table 4 Performance comparison of junior dentists, machine learning, and machine learning-assisted junior dentists in demineralization identification on the test set (see Table S1 for detailed dentition test results). AI, artificial intelligence.

	Operator	TP	FP	TN	FN	NPV	PPV	SEN	SPEC	F1-score	
Individual	Junior 1	Manual	12	2	6	0	1.000	0.857	1.000	0.750	0.923	
DL-assisted	11	0	8	1	0.889	1.000	0.917	1.000	0.957	
Junior 2	Manual	12	3	5	0	1.000	0.800	1.000	0.625	0.889	
DL-assisted	12	0	8	0	1.000	1.000	1.000	1.000	1.000	
AI		11	0	8	1	0.889	1.000	0.917	1.000	0.957	
Photo	Junior 1	Manual	30	7	23	0	1.000	0.811	1.000	0.767	0.896	
DL-assisted	28	0	30	2	0.938	1.000	0.933	1.000	0.966	
Junior 2	Manual	30	11	19	0	1.000	0.732	1.000	0.633	0.845	
DL-assisted	29	0	30	1	0.968	1.000	0.967	1.000	0.983	
AI		28	2	28	2	0.933	0.933	0.933	0.933	0.933	
Teeth	Junior 1	Manual	62	47	368	3	0.992	0.569	0.954	0.887	0.713	
DL-assisted	65	15	400	0	1.000	0.813	1.000	0.964	0.897	
Junior 2	Manual	63	55	360	2	0.994	0.534	0.969	0.867	0.689	
DL-assisted	65	7	408	0	1.000	0.903	1.000	0.983	0.949	
AI		60	26	389	5	0.987	0.698	0.923	0.937	0.795	

Figure 4 (A–I) Performance of deep learning models in detecting enamel demineralization.

(A–C) Patient retained photo, (D–F) artificial intelligence (AI) assisted labeled chart of a junior dentist, (G–I) the gold standard.

Comparison of DL-assisted and dentist-only detection

We contrasted the annotation performance and efficiency of junior dentists with and without DL assistance in detecting demineralization on digital pictures to evaluate the model’s applicability thoroughly. Our data show a significant improvement in the performance of junior dentists with DL assistance compared to without, both on a per-image and per-tooth basis: Dentist 1 (Images: p = 0.099; Teeth: p = 0.019), Dentist 2 (Images: p = 0.026; Teeth: p < 0.01). The detection capability of two junior dentists for dental enamel demineralization improved significantly, with the F1-scores for demineralized teeth increasing from 0.713/0.689 to 0.897/0.949 respectively. The performance in SEN and SPEC was equally remarkable. However, the standalone model and junior dentists without assistance demonstrated lower sensitivity and higher specificity in demineralization identification. The primary challenge for junior dentists was the tendency to misclassify, which was effectively addressed with DL assistance, outperforming either of the two alone.

In addition, with the help of DL, two junior dentists made satisfactory improvements in the identification of demineralized areas. Regarding the demineralization area, the accuracy of their annotations improved with DL assistance, as indicated by the Intersection over Union (IoU). Their IoU scores increased from 0.571/0.527 to 0.704/0.682, respectively.

Basis of demineralization index

Figure 5 shows that the tooth contours outlined by the DL model closely match the real contours. Based on the test set, the IoU of the tooth contours with the gold standard reached 90.3%. This allows the DL-drawn contour area to be directly used as the denominator when calculating the percentage of demineralized area for individual teeth. Thus, this lays the foundation for the automated calculation of the demineralized index. We found that the average demineralization index was highest for upper molars (0.243) and lowest for lower incisors (0.104), precisely in line with clinical patterns of caries occurrence. This indicates the index’s practical value to some extent.

Figure 5 Comparison of machine-drawn tooth contours and contours annotated by senior dentists.

(A) Machine-recognized tooth contours, (B) tooth contours annotated by senior dentists.

Discussion

Early and accurate detection of the WSL is a crucial factor in preventing and treating dental caries. However, early lesions are often subtle and challenging to detect due to minimal progression. The current challenges include relatively rough grading and the energy and time required for continuous detection. Failure to detect and monitor enamel demineralization can rapidly progress to extensive caries, particularly for patients undergoing orthodontic treatment. Therefore, automated monitoring is necessary to alleviate the burden and enhance management. DL models, particularly CNN, have demonstrated significant potential in dental image analysis. The study by Schönewolf et al. (2022), which also utilized a CNN for the automated detection and classification of molar-incisor hypomineralization, has illustrated the high-precision segmentation capabilities of DL models for enamel demineralization areas on tooth surfaces. Besides, the advantages also include automation and high efficiency (Neumayr et al., 2024), as well as aiding in clinical decision-making, such as enhancing the diagnostic capabilities of junior dentists as demonstrated in our study. We utilized DL and digital photos to automatically detect and quantify the degree of enamel demineralization on the tooth surface especially during orthodontic treatment.

The results of the study demonstrate that the automated approach using a DL model can accurately assess the degree of enamel demineralization on tooth surfaces. Compared to the manual methods of senior dentists, the fully automated method is more efficient. With the assistance of the DL model, junior dentists improved their accuracy in detecting enamel demineralization on tooth surfaces. With DL assistance, junior dentists can effectively address issues related to their lack of experience, such as misidentification and overlooking subtle demineralization areas. The recognition ratio of the demineralization area reached 0.852/0.829, comparable to the level of recognition by senior dentists. DL excels at diagnosing and annotating enamel demineralization on tooth surfaces, improving the diagnostic accuracy, precision, and efficiency of dentists in detecting enamel demineralization.

We also gave a possible refined demineralization grading which imitates the plaque index, as shown in Fig. 6 and Table 5. The DL model effectively recognizes tooth contours and then calculates annotated areas. The calculation of the demineralization index makes the progression of the disease more quantified and monitored and can detect the small lesions more accurately. Comparing the indexes at each time point after fixing the digital photo position allows for improved assessment, reflecting the progression of demineralization. The model provides a dynamic monitoring solution for early enamel demineralization. By quantifying the degree of demineralization and specifying clinical diagnoses, patients’ demineralization status can be accurately monitored, thereby reducing the work burdens of dentists.

Figure 6 Schematic of optimized demineralization classification.

Table 5 Scoring criteria of a possible grading of the enamel demineralization.

Grade	Criteria	
0	No demineralization on the tooth surface	
1	Demineralized area coverage less than 1/3 of the tooth surface	
2	Demineralized area coverage between 1/3 and 2/3 of the tooth surface	
3	Demineralized area coverage exceeding 2/3 of the tooth surface or caries	

However, the study has some limitations. The study was based on a single institutional source and involved a small sample size. Due to the shot angles, upper and lower anterior teeth were counted from one frontal photograph, while upper and lower posterior teeth were counted from two lateral occlusal photographs. This method does not fully account for dental arch morphology. As a result, it is relatively crude and less accurate, especially for patients with malocclusion and missing teeth, reducing its applicability. Additionally, calculating the ratio of demineralized area to tooth surface area can be challenging due to factors such as deep overbite, overlap, or gingival recession. This can decrease the reliability of the index. However, these shortcomings do not diminish its role as an indicator of tooth demineralization severity, which is crucial for early caries lesion grading and continuous monitoring.

Further research and application of this method can improve the accurate detection and grading of early enamel demineralization. This can facilitate early management of enamel demineralization and provide more accurate treatment plans for patients. Using digital photographs taken from a fixed angle is an efficient and non-invasive method. The model’s calculation of the demineralization index enables continuous monitoring, especially for orthodontic patients, helping prevent further demineralization and extensive caries. The study results show that our DL model can also assist in the education of junior dentists, enhancing their ability to recognize and judge enamel demineralization. Future studies could focus on and validate the model’s ultimate ability for disease progression to solidify its clinical applicability. In addition, it is necessary to extend this dataset to multiple institutions, as well as to use multimodal model data, such as X-ray rays, 3D scans or patient history.

Conclusions

The application of DL significantly enhances the accuracy, precision, and consistency of diagnosing and annotating dental enamel demineralization. DL effectively assists junior dentists in assessing demineralization areas. The index can quantify demineralization severity, by calculating the proportion of demineralized area, aiding in early classification and caries prevention. By incorporating artificial intelligence (AI) into routine dental exams, patient care can be improved, reducing the risk of dental caries occurring and promoting better oral health outcomes.

Supplemental Information

Supplemental Information 1 Recognition of each tooth position by the test set.

Supplemental Information 2 Recognition of each tooth position by the validation set.

Supplemental Information 3 Demineralization index of each tooth position.

Supplemental Information 4 Codebook.

Supplemental Information 5 Model-related code.

Supplemental Information 6 IoU of the teeth.

Supplemental Information 7 Raw data: Characteristics of the patients.

Supplemental Information 8 The recognition of various images and tooth positions by junior dentists with and without DL assistance (the data of the AI group is directly generated to obtain the results we need).

We thank Mr. Tian Erkang, for his valuable suggestions on the outline of this article, journal selection, and other aspects.

Additional Information and Declarations

Competing Interests

Author Contributions

Human Ethics

Data Availability

The authors declare that they have no competing interests. Ruoshi Chen and Jianpeng Ren are employed by Chengdu Boltzmann Intelligence Technology Co., Ltd.

Ketai He conceived and designed the experiments, performed the experiments, prepared figures and/or tables, and approved the final draft.

Rongxiu Zhang performed the experiments, authored or reviewed drafts of the article, and approved the final draft.

Muchun Liang performed the experiments, prepared figures and/or tables, and approved the final draft.

Keyue Tian performed the experiments, prepared figures and/or tables, and approved the final draft.

Kaihui Luo performed the experiments, prepared figures and/or tables, and approved the final draft.

Ruoshi Chen analyzed the data, prepared figures and/or tables, and approved the final draft.

Jianpeng Ren analyzed the data, prepared figures and/or tables, and approved the final draft.

Jiajun Wang analyzed the data, prepared figures and/or tables, and approved the final draft.

Juan Li conceived and designed the experiments, authored or reviewed drafts of the article, and approved the final draft.

The following information was supplied relating to ethical approvals (i.e., approving body and any reference numbers):

The West China Hospital of Stomatology Ethics Committee.

The following information was supplied regarding data availability:

The raw data and code are available in the Supplemental Files.

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
