# Peer review of "The application of deep learning in early enamel demineralization detection"

_PeerJ, doi:10.7717/peerj.18593_

## Round 0.1 · original submission · Major Revisions

Your paper has been reviewed by three reviewers. All of them see potential but there are major issues that should be addressed before your paper could be considered for publication. Besides the other issues as pointed out by the reviewers, especailly the rather small sample size is a methodological issue in the development of the model. Furthermore, I may have missed it, but I cannot find a certified translation of ethical approval, there is only a copy of a letter in Chinese. Also, limitations of the study shouold be discussed in the discussion section.

Please address all remarks of the reviewers, point by point, in your rebuttal letter. Repeat a remark, and give your response below each remark.

·

Basic reporting

Abstract:
1. Objective: kindly delete the first sentence, and instead of our objectives, start with the current study objectives
2. Methods: write about deep learning model details
3. Results: Kindly provide significant values
4. Keywords are missing
Introduction:
1. Write briefly about the deep learning models in dentistry
2. Provide the literature on the same
3. Briefly write on the justification of the current study
4. Importance of digital photographs and caries diagnosis

Experimental design

Materials and methods:
1. A total of 237 adult patients: Did the authors calculate the sample size? Please provide details and justification for the same. Were these number of samples enough to address the objectives of the current study
2. Two senior dentists reached a consensus through clinical examinations and patient history discussions, marking tooth contours and demineralized lesions on the photos, which served as the required samples for the study: The two senior dentists are the current study's authors? Please provide the calibration of the examiners.
3. Dental photos were taken: who took them?
4. Two senior dentists, two junior dentists, and one computer scientist: same as point 2
5. Labelme: Kindly provide details like version, ownership, etc
6. QT, JSON? Please provide the complete form

Validity of the findings

Results:
1. The comparative analysis is not clear
2. Expertise outcome between senior and junior dentist
3. Diagnosis comparisons
4. Please provide the significant p-values

Additional comments

Discussion:
1. DL model can accurately assess the degree of enamel demineralization on tooth surfaces: discuss this with the literature.
2. DL is superior in diagnosing and annotating enamel demineralization on tooth surfaces: justify writing in detail with suitable references.
3. The last paragraph is the conclusion, and kindly delete the same
4. Kindly provide all the limitations of the current study
5. Write briefly on the clinical implementation, dental education and dental practice
6. Recommendation per the outcome of the current study

Reviewer 2 ·

Basic reporting

-Ensure sentences are clear and concise.
-Avoid unnecessary repetition and overly complex sentence structures.
-Ensure subject-verb agreement and proper verb tense usage.
-Use active voice where possible to make sentences more direct and dynamic.
-Use commas to separate clauses for better readability.
-Ensure consistent use of periods and commas within lists and complex sentences.
-Break down long sentences into shorter, more manageable ones.
-Use transitional phrases to improve the flow between ideas and sections.
-Carefully proofread the manuscript to catch any typographical errors or inconsistencies.
-Consider using grammar and spell-check Editors to assist with identifying and correcting errors.
-Ensure logical transitions between paragraphs and ideas.
-Remove redundant information and focus on clear, concise statements.
-Ensure all citations are appropriately formatted and consistently presented.

Experimental design

The manuscript "The application of artificial intelligence in early enamel demineralization detection" has been submitted to PeerJ.

The purpose of this research was to create a diagnostic model utilizing intraoral photographs to accurately identify, classify, and detect enamel demineralization at an early stage on tooth surfaces. The study's findings indicate that deep learning techniques can precisely segment tooth surfaces and lesion boundaries, thereby improving the accuracy, precision, and efficiency of diagnosing enamel demineralization and delineating affected areas.

While the manuscript tackles an important issue, there are several concerns regarding the study.

Specific comments are noted below:

Title: Indicate the type of study. Consider also adding the key methodology (deep learning) to the title for clarity and specificity.

Abstract
- The objective statement is clear but can be made more concise. Suggested revision: "This study aimed to develop a diagnostic model using intraoral photographs for accurate identification, classification, and early detection of enamel demineralization on tooth surfaces."
- Clarify the nature of the study design (e.g., observational, experimental).
- Provide more details about the patient population, such as age range or demographics, if relevant.
- Specify the type of deep learning model used.
- Break down the results into clearer points and use consistent formatting.
- Explain the significance of the F1-score for readers who may not be familiar with this metric.
- Clarify the comparative performance results for easier understanding.
- Summarize the implications of the findings clearly and concisely.

Keywords: Please ensure that all of them correspond to MeSH terms.

Introduction
- Lines 43-46: The initial definition is clear, but you could streamline it for conciseness. Consider combining related information for a smoother flow.
- Lines 47-51: Clarify the causative factors and consequences in a more concise manner.
- Lines 51-58: Enhance clarity and focus on the classification details. Clarify the challenges of early detection.
- Lines 59-67: Provide a smoother transition to the current clinical detection challenges.
- Lines 68-75: Highlight the potential benefits of deep learning clearly.
- Lines 76-90: Ensure clarity and logical flow in presenting the study's aim and methodology.
- Highlight the novelty and significance of the study.
- Ensure logical transitions between paragraphs and ideas.
- Remove redundant information and focus on clear, concise statements.
- Ensure all citations are appropriately formatted and consistently presented.

M&M
- Lines 93-106: Clarify the study period and the rationale for selecting specific patient criteria. Ensure consistent formatting for author names and citation styles.
- Lines 107-110: Clearly state inclusion and exclusion criteria.
- Lines 111-117: Improve clarity and detail of the experimental setup.
- Lines 118-121: Clarify the methodology and provide context for the use of orthopantomography.
- Lines 122-134: Streamline the description of the annotation tool and process. Emphasize the consistency and reliability of annotations.
- Lines 135-141: Ensure clarity and detail in the preprocessing steps.
- Lines 142-155: Clearly describe the data augmentation techniques and model architecture.
- Lines 158-175: Clarify and define the metrics used for evaluation.
- Lines 176-185: Clearly outline the evaluation process and consistency checks.
- Lines 186-194: Provide detailed technical specifications for reproducibility.
- Clearly outline each step and ensure the methodology is easy to follow.
- Provide sufficient detail for reproducibility, especially in technical and procedural descriptions.
- Ensure all citations are correctly formatted and consistently presented.

Validity of the findings

Results
- Lines 197-205: Ensure clarity and detail when discussing model performance and metrics. Use full terms before abbreviations when first introduced.
- Lines 206-214: Provide context for the comparison and elaborate on specific performance metrics.
- Lines 215-230: Clearly describe the comparative analysis and emphasize the improvements with DL assistance. Ensure consistency in terminology and metrics.
- Clearly outline each result and ensure the methodology is easy to follow.
- Provide sufficient detail for reproducibility, especially in technical and procedural descriptions.
- Ensure all figures and tables are correctly referenced and provide meaningful insights.

Discussion
- Lines 233-244: Start by emphasizing the importance of early detection and the challenges involved. Clearly state the study’s aim and its potential impact.
- Lines 245-256: Highlight the key findings and their implications. Compare the performance of the DL model to traditional methods.
- Lines 257-272: Emphasize the advantages of using DL for diagnosing and annotating enamel demineralization. Mention the potential of the demineralization index for clinical use.
- Lines 273-277: Discuss the enhancements in monitoring and assessment. Highlight the efficiency and accuracy of the DL model.
- Lines 278-290: Address the limitations of the study and suggest future research directions.
- Lines 291-297: This paragraph belongs to the conclusions described in the following paragraph; therefore, it should be omitted.

Conclusions
- Summarize the main findings and their clinical implications.
- Highlight the potential for further research and application.

·

Basic reporting

1. The abstract lacks a conclusion summarizing the main findings and their implications. This would help readers understand the significance of the study.
2. Objective: The abstract mentions the development of a diagnostic model based on intraoral photographs but does not specify the type or architecture of the deep learning model used. This is crucial for understanding the study's novelty and technical depth.

Experimental design

The study involved 208 patients from a single institution. This raises concerns about the sample size and distribution.
A single institution might not represent a diverse patient population, potentially limiting the model's applicability to broader populations.
There is no mention of specific criteria for patient selection or the standard conditions under which the images were captured. This lack of detail can make it difficult to assess the consistency and quality of the data.
It mentions the use of 624 high-quality digital images but does not clarify how these images were divided into training and testing sets. Proper validation techniques, such as cross-validation, should be discussed to ensure the model's sturdiness.

Validity of the findings

The recognition and segmentation of tooth contours are described as "visually excellent," but this is a subjective measure. Objective metrics should be provided to substantiate this claim.
The concept of a demineralization index is introduced, but the abstract does not explain how this index is calculated or validated.
The F1-score for identifying demineralized teeth is reported as 0.856. While this is a good score, it would be more informative to provide additional metrics such as precision, recall, and the area under the receiver operating characteristic (ROC) curve to give a complete picture of the model's performance.
The performance improvement of junior dentists with the assistance of deep learning is noted, but it would be useful to know the baseline performance of senior dentists for comparison.
Additionally the study does not clarify if the junior dentists' performance was evaluated on the same test set or in a clinical setting.
Ethical considerations and patient consent are not mentioned. This is crucial for studies involving human subjects, especially in medical research.
The results mention average F1-scores for demineralized tooth identification increasing significantly but do not discuss potential overfitting of these improvements.
There is also no mention of statistical significance testing to confirm that the improvements are not due to chance.
Provide detailed information on the deep learning model architecture and the training/testing split of the data.

Additional comments

Include objective metrics and statistical analysis to substantiate claims about model performance.
Discuss the potential limitations regarding sample size, single-institution data.
Clarify the calculation and validation of the demineralization index.
Add a conclusion summarizing the study's significance and implications.
Mention ethical considerations and patient consent.

---

## Round 0.2 · Major Revisions

While you addressed most of the comments, there are still important issues raised by Reviewer 3 that require your attention. This reviewer provided valuable feedback that will strengthen your paper. Additionally, the manuscript's flow could benefit from consulting a professional academic copyediting service.

·

Basic reporting

None

Experimental design

None

Validity of the findings

None

Additional comments

The authors have addressed all the comments and suggestions, and the manuscript has dramatically improved. I suggest that the manuscript can be accepted for publication in its current form. I want to congratulate the authors and wish them the best in their future endeavours.

Best regards and keep well

Reviewer 2 ·

Basic reporting

The authors made the suggested changes

Experimental design

The authors made the suggested changes

Validity of the findings

The authors made the suggested changes

Additional comments

None

·

Basic reporting

The flow in which the abstract is written needs more polishing in terms of could be improved for clarity.
There are some fragmented sentences and abrupt transitions that make the text harder to follow
The wording in Results is also confusing in places

Experimental design

The explanation of the performance metrics (F1-scores) could be clearer. The average F1-scores for junior dentists has some redundancy, and the results could be presented more straightforwardly
There is little detail on how the "demineralization index" was calculated or implemented, though it is mentioned multiple times. If space allows a brief explanation would help clarify this novel aspect of the study
Several grammatical errors and awkward phrasing undermine from the readability many sentences should be rewritten for clarity
A consistent tense and voice would also improve readability, introduction moves between passive and active voice sometimes awkwardly. Consistency here would make the text smoother
The introduction jumps between topics, sometimes interrupting the logical flow also to mention certain phrases are repetitive
The use of technical terms is sometimes inconsistent. For instance the introduction alternates between terms like “demineralization” “white spot lesion (WSL)” and “cavitation”, without clearly distinguishing them for readers unfamiliar with these terminologies
The demineralization index is introduced as a key aspect of the study but more detail on how it works and its importance in comparison to previous grading systems (e.g the WSL grading system) would be beneficial

Validity of the findings

The section where the authors transition into discussing DL and digital photography is crucial but lacks a smooth lead-in. It would be more effective to first describe the shortcomings of current detection methods and then introduce DL as a solution
Consider reorganizing the introduction into distinct sections:
(a) Clinical Relevance of Enamel Demineralization
(b) Current Detection Methods and Their Limitations and
(c) Advancements in Deep Learning for Dental Diagnostics
This structure would create a more coherent narrative, guiding the reader smoothly from problem to solution
More detailed discussion on the demineralization index would add clarity. How does this new index improve diagnostic accuracy? How is it calculated, and why is it a better measure than previous systems?
Larger Dataset: A larger and more diverse dataset would help the model generalize better. Currently, the model is trained on 624 images from a single institution. Expanding this dataset across multiple institutions, with a broader range of patients and clinical settings, would reduce overfitting and improve the model's ability to detect different demineralization patterns.
Multi model Data: Using other data types such as X-rays, 3D scans or patient history could provide a richer set of inputs for the model enabling better context-based decision-making
Scope for further: validation through longitudinal studies that test the models ability to track disease progression over time, particularly in orthodontic patients would solidify its clinical applicability

Additional comments

Please make the above corrections and revert back with all the necessary improvements in the manuscript

---

## Round 0.3 · accepted · Accept

I have re-assessed the second revision of your manuscript. Thank you for meeting the comments of reviewer 3. Congratulations with your fine work, the paper is now ready for publication.